# Effect of Heat Treatment on the Resulting Dimensional Characteristics of the C45 Carbon Steel after Turning

Jana Moravčíková [1], Roman Moravčík [2] and Marián Palcut [2,*]

1   Institute of Production Technologies, Faculty of Materials Science and Technology in Trnava, Slovak University of Technology, 917 24 Trnava, Slovakia
2   Institute of Materials Science, Faculty of Materials Science and Technology in Trnava, Slovak University of Technology, 917 24 Trnava, Slovakia
*   Correspondence: marian.palcut@stuba.sk

**Abstract:** The presented article deals with the influence of the heat treatment of C45 steel on the surface quality after turning. Turning is a machining technology used to prepare specific geometrical characteristics of surface and dimensional quality. In the present paper, the same turning conditions were used for the turning treatment of differently heat-treated steels. The soft annealed state, normalized state, hardened state, quenched and tempered at 530 °C state and quenched and tempered at 660 °C state have been analyzed. By using this approach, it has been possible to evaluate the effects of hardening and machining on the resulting parameters after turning (roughness, cylindricity and circularity). The highest hardness was observed in the steel after the hardening process (694 ± 9 HV 10). The hardening has negatively influenced the surface quality. The high hardness was related to martensite formation and caused a damage to the cutting edge of the cut insert used, leading to a significant change in geometrical accuracy. The cylindricity change achieved 0.15 ± 0.03 mm which was significantly higher compared to the theoretical value of the diameter of the machined steel bar. An inaccuracy was also observed in diameter dimension. These inaccuracies were caused by the wear of the cutting edge of the tool used in cutting parameters setting.

**Keywords:** C45 steel; heat treatment; turning; roughness; roundness; microstructure





## 1. Introduction

Turning is one of the basic ways of materials machining. It is used to produce technical parts with a defined geometry and dimensional accuracy. The machinability is an important characteristic of materials [1–3]. The accompanying phenomenon of machinability is the dimensional accuracy and roughness achieved during machining. Furthermore, the material of the cutting wedge, and the type of coating also influence the resulting surface parameters [4–7]. In technical practice, the machining of semi-finished products is streamlined to ensure the efficiency and economy of production. In the metal cutting a pressing of a cutting tool against the workpiece is involved [8]. The pressing is done with a certain degree of external force and results in a material removal from the workpiece in the form of chips. In the metal cutting process, the tool performs the cutting action by overcoming the shear strength of the workpiece [8]. This action generates a large amount of heat in the workpiece, resulting in a highly localized and thermo-mechanically coupled deformation in the affected zone [8,9]. Metal cutting can be associated with high temperatures in the tool-chip interface zone. The thermal aspects of the cutting process may affect the accuracy of the machining process. The high cutting temperatures influence the tool wear rate, its lifespan, surface integrity of the workpiece and chip formation mechanism [9]. Furthermore, the generated heat may contribute to the thermal deformation of the cutting tool. The increased temperature of the workpiece in the primary deformation zone softens the material. As such, it decreases the cutting forces and reduces the energy required to cause further shear. The processed semi-finished products are expected to be delivered in a

certain initial state. The C45 steel (AISI 1045) is delivered in the state after normalization annealing [10]. During machining, however, the material may experience a significant heat generation due to friction. The increased temperature at the tool-chip interface affects the contact phenomena by changing the friction conditions, which in turn influence the location and shape of the primary and secondary deformation zones, maximum temperature, heat dissipation and diffusion of the tool material into the chip [9]. Furthermore, the hardness, microstructure and chemical composition of a workpiece material have a significant effect on the main cutting force, and on the resulting cutting energy [11,12]. Therefore, the aim of the present paper is to determine the influence of the heat treatment on the resulting geometric and micro-geometric characteristics of the tool during turning.

The C45 steel is defined as a standard material for machinability in the state after normalization annealing [10]. Since the C45 steel was defined as an etalon for steel machining, many authors have analyzed different ways of machining and its influence on the dimension accuracy and resulting machined surface roughness [13–17]. It has been found that the grinding of the C45 steel causes a work hardening of its surface layer without phase transformation [17]. A nanoindentation test on the cross-section, performed at a short distance from the grinded surface, has shown that ferrite grains were more susceptible to work hardening than pearlite grains due to the creation of an equiaxed cellular microstructure [17]. A different dislocation substructure was created in the work-hardened surface layer after grinding in different depths. Furthermore, the cooling during machining has also been found to be an important parameter required to achieve the desired surface quality [18]. Optimized dry turning or turning with minimum lubrication can be used to provide a surface roughness and other parameters that are comparable to traditional emulsion cooling [18]. By using this approach, it is possible to minimize manufacturing costs and reduce environmental hazards related to the use of various lubricants. Bouchareb et al. studied the effect of depth of cut on vibration amplitude during rough face milling of C45 steel on CNC machine [19]. It has been found that the height to width ratio should stay below 1.5 to avoid significant vibration amplitudes. Several authors have studied the surface integrity, and the influence of cutting material and cutting angle on the quality of the machined surface [20–22]. It has been found that the initial state of the material may affect the surface roughness [23–25], but also the lifetime of the cutting tool used [26]. A cryogenic treatment improves the surface roughness by decreasing the retained austenite matrix to martensite [27]. The hardening also influences the parameters of orthogonal cutting [28].

The geometry of the cutting wedge is an important factor that influences the precision and quality of the machined surface [29]. The roughness of the machined surface is created by copying the cutting wedge of the cutting tool used during machining, depending on the cutting parameters and conditions used. Another important factor is the state of the material during machining. The change of microstructure caused by heat treatment causes a change in hardness. It is a result of a change in the distribution and morphology of the phases present in the material, and thus has a significant influence on the cutting parameters, which should be optimal for the given condition [30]. A significant change in hardness causes a higher resistance of the material to the penetration of the cutting wedge into the material, and thus, at the same cutting parameters, the wear of the cutting wedge (durability and tool-life) may be different, e.g., lower. The durability of the tool can also be limited by different wear rates [31,32]. One can talk about the durability in terms of wear of the minor or major flank, face, allowable roughness, etc. Therefore, the increase of the hardness has also a certain effect on the shape tolerances, such as, e.g., cylindricality and roundness. If cutting forces or surface roughness are limited by predetermined values, it is not possible to allow any wear to the cutting wedge [33,34]. As such, there is a certain limit wear of the allowable cutting force and surface roughness. It is necessary to know the limit wear to determine the optimal wear of the cutting wedge.

The application of heat treatment is important in various areas of metal production, including additive manufacturing of semi-finished products, which are subsequently ma-

chined to final dimensions [35,36]. Additive manufacturing can be regarded as a disruptive process that builds complex components layer by layer [37]. Selective laser melting and electron beam melting are powder bed fusion processes that are frequently used in additive manufacturing [38,39]. These processes are used to manufacture metallic parts with the aid of a beam. Several authors investigated the additive manufacturing of metal parts from different perspectives and on different materials [39,40]. It has been found that heat treatments may modify the microstructure, reduce residual stresses, and increase the ductility, fatigue life, and hardness of the components [41,42]. For example, a 17-4 PH steel (EN 1.4542, AISI 630) exhibits a dendritic structure when fabricated via additive manufacturing [42]. The microstructure of additively manufactured 17-4 PH contains a large fraction of nearly retained austenite along with body centered cubic/martensite and fine niobium carbides preferentially located along inter-dendritic boundaries. By implementing the recommended homogenization heat treatment, the dendritic solidification microstructure can be eliminated, thereby resulting in a microstructure containing about 90% martensite with 10% retained austenite [42].

The current article deals with the influence of the heat treatment of C45 steel on the surface quality after turning. Identical turning conditions have been used for the turning treatment of differently heat-treated steels. The soft annealed state, normalized state, hardened state, quenched and tempered at 530 °C state and quenched and tempered at 660 °C state have been analyzed. These states could lead to a damage (wear) of the active part of the tool in the process of turning, under precisely defined conditions. Based on the recommended temperatures for heat treatments listed in [43–45], the individual temperatures of the implemented heat treatment methods were determined. Due to the dimensions of the bar semi-finished product used (Ø30 × 100 mm), the holding time at individual temperatures was set at 1 h. This time should be sufficient to achieve homogeneous austenitization in the entire cross-section during high temperature heat treatments and result in the desired changes. The present work is novel as it investigates all possible states of C45 steel. To our best knowledge, the effect of heat treatment of C45 steel on the resulting surface quality after turning has not been investigated yet.

## 2. Materials and Methods

C45 is a medium carbon steel that can be used when greater strength and hardness are required compared to the "as rolled" condition. An extreme size accuracy, straightness, and concentricity may be combined to minimize wear in high-speed applications. The steel can be turned, ground, and polished. Quenched and subsequently tempered steel can be used for screws, forgings, wheel tires, shafts, sickles, axes, knives, woodworking drills, hammers, etc. [44,45]. The C45 carbon steel (1.0503, ASTM 1045) was used in the form of a round bar. The nominal [43] and experimentally determined chemical composition of the steel is shown in Table 1. The chemical composition was measured 5-times on a Bruker Emission Spectrometer, type Q4 Tasman.

**Table 1.** Nominal [43] and actual chemical composition (wt. %) of C45 carbon steel used in this work.

| | Nominal [43] | | Actual |
|---|---|---|---|
| | Min. (wt.%) | Max. (wt.%) | |
| C | 0.42 | 0.50 | 0.487 ± 0.005 |
| Cr | - | 0.40 | 0.074 ± 0.001 |
| Mn | 0.50 | 0.80 | 0.718 ± 0.005 |
| Ni | - | 0.40 | 0.052 ± 0.003 |
| P | - | 0.045 | 0.0098 ± 0.0002 |
| S | - | 0.045 | 0.011 ± 0.001 |
| Si | - | 0.40 | 0.158 ± 0.008 |
| Mo | - | 0.10 | 0.017 ± 0.001 |
| Cr + Mo + Ni | - | 0.63 | 0.143 |

For the analysis, a C45 steel bar with a diameter of 30 mm and length of 100 mm was used. Heat treatments were carried out in a muffle furnace under a protective $N_2$ atmosphere. The following heat treatment types were used: soft annealing, normalizing, hardening, quenching, and tempering. The treatments were performed at low and high temperatures to achieve the quenched and tempered state using the low or high tempering. The parameters of the heat treatments are shown in Table 2. The hardened steel was machined immediately after the hardening process.

**Table 2.** Heat treatment parameters of C45 steel.

| Steel Number | Heat Treatment Type |
|---|---|
| 1 | Soft annealing at 700 °C/1 h holding time, slow cooling in $N_2$ for 10 h |
| 2 | Normalizing at 850 °C/1 h holding time, cooling in air |
| 3 | Hardening at 830 °C/1 h holding time, cooling in water |
| 4 | Quenching and tempering followed by hardening at 830 °C/1 h holding time, cooling in water; tempering 530 °C/1 h holding time and cooling in air |
| 5 | Quenching and tempering followed by hardening at 830 °C/1 h holding time, cooling in water; tempering 660 °C/1 h holding time and cooling into air |

After heat treatment, the steels were wet ground using sandpapers (grades 60, 120, 240, 400, 600, 1200) and polished by Buehler Metadi Supreme diamond suspensions (9, 6, 3 and 1 μm). For etching the 3% Nital (3% of $HNO_3$ in ethyl-alcohol) was used. The resulting microstructures were observed and documented by a light microscope, type Neophot 32, with an Olympus E-420 digital camera installed and operated by Quick Photo Media software.

For studying the effect of heat treatments on hardness, a Vickers hardness test was performed according to EN ISO 6507-1 standard. We used the Zwick 3212 device with 98.07 N loading for indentation.

A lathe SUI 500 COMBI was used for machining. The lathe can be used for conventional machining or machining controlled with a CNC program. The Korloy WWLNR2525-M08 toolholder type and the Korloy WNMG080404-HM NC3020 cutting insert type were employed. The recommended cutting parameters used for the selected tool are listed in Table 3 [46]. All heat-treated samples were machined under the same conditions, given in Table 4. The machining geometry is schematically shown in Figure 1.

**Table 3.** Recommended cutting parameters used for the selected tool [46].

| Cutting Parameter | Value |
|---|---|
| Cutting speed $v_c$ [m·min$^{-1}$] | $180 \div 300$ |
| Cutting depth $a_p$ [mm] | $1 \div 5$ |
| Feed f [mm] | $0.1 \div 0.5$ |

**Table 4.** Cutting parameters for machining of heat-treated samples.

| Operation | Description | Cutting Depth $a_p$ [mm] | Feed f [mm] | Cutting Speed $v_c$ [m·min$^{-1}$] |
|---|---|---|---|---|
| 1 | Facing 1 mm | 1 | - | - |
| 2 | Turning Ø30–30 mm | 1.5 | 0.2 | 180 |
| 3 | Turning Ø27–30 mm | 1 | 0.2 | 180 |
| 4 | Edge machining | 0.5 × 80° | - | - |
| 5 | Rotating the part | | | |
| 6 | Facing 1 mm | 1 | - | - |
| 7 | Turning Ø30–40 mm | 1.5 | 0.2 | 180 |
| 8 | Turning Ø27–40 mm | 1 | 0.2 | 180 |
| 9 | Edge machining | 0.5 × 80° | - | - |

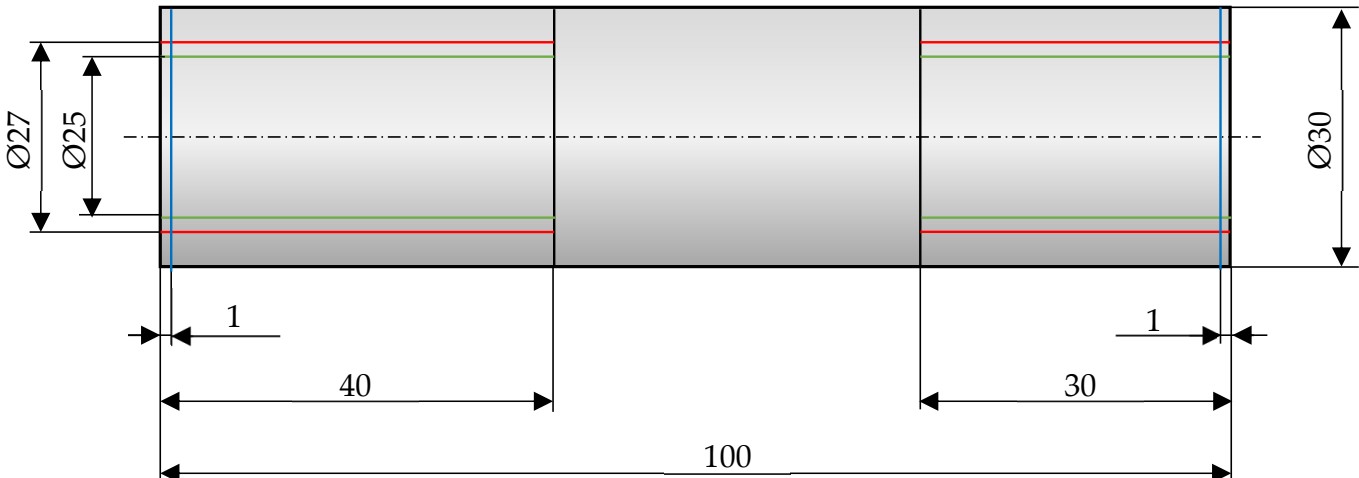

**Figure 1.** Scheme of machining of the steel bar according to cutting parameters given in Table 4. Dimensions are given in millimeters.

The machined surface parameters of each steel were measured on the Surfcom 5000 device (Accretech Tokyo Seimitsu, Japan) using the stylus Accretech Tokyo Seimitsu DM48515 with a radius of 2 μm, according to the standard ISO 97/09 with a measured length of 4 mm. The measurements were carried out three times on each machined surface on different places.

The cylindricity and circularity (diameter sizes) on each machined surface of the samples were determined next. The measurement was performed using the Carl Zeiss QEC GmBH Prismo®Ultra device (Peine, Germany) on both machined surfaces of the heat-treated samples. After machining each sample, the state of the cutting edges of the used cutting insert was controlled and evaluated by using the Carl Zeiss QEC GmBH Stemi 2000 stereomicroscope (Peine, Germany).

## 3. Results and Discussion

This section details the results and discussion of individual measuring tests of heat treated and machined C45 steel. The microstructure, hardness, surface roughness, cylindricality, and diameter size changes are presented in separate subchapters.

### 3.1. Microstructure of the Analyzed Samples

Figure 2 documents the microstructures of the heat-treated samples. The microstructure of the sample in the soft-annealed state is given in Figure 2a. Ferrite areas (white grains) and spheroidized pearlite can be observed. The original austenitic grain size was determined by comparative methods according to the EN ISO 643 standard and a value of 35 μm was found. Figure 2b documents the microstructure of the sample after normalizing. The grain size of the original austenite was smaller (approximately 14 μm) compared to the previous microstructure type. The microstructure consisted of ferrite and lamellar pearlite. The amount of ferrite was larger compared to the state after soft annealing. The microstructure in Figure 2c documents the sample after hardening. The microstructure consisted of martensite, retained austenite, and a small amount of troostite. The microstructure in Figure 2d is for the quenched and tempered at 530 °C state. It consists of tempered martensite and precipitated cementite particles (low tempered martensite). After quenching and tempering at 660 °C the microstructure consisted of sorbite (a very fine mixture of ferrite and cementite particles, Figure 2e). The original austenitic grain size was not determined because of the very fine microstructure types of the samples treated by hardening and quenching.

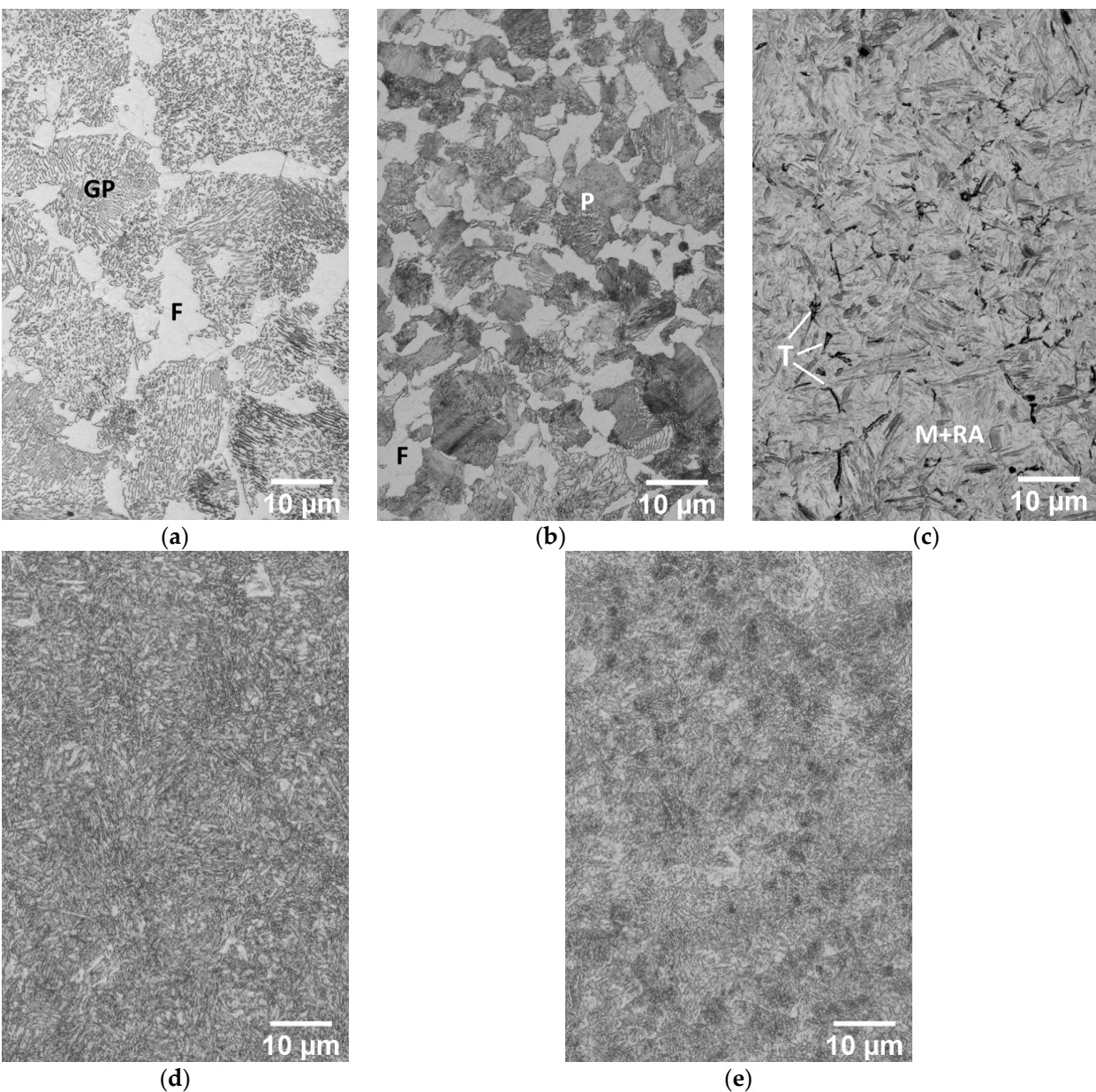

**Figure 2.** Microstructures of heat-treated samples: (**a**) soft annealed; (**b**) normalized; (**c**) hardened; (**d**) quenched and tempered at 530 °C; (**e**) quenched and tempered at 660 °C (labeling: GP = grained pearlite; F = Ferrite; P = pearlite; T = troostite; M = martensite; RA = retained austenite).

### 3.2. Vickers Hardness Test

The hardness was measured on the cross section for each treated steel according to EN ISO 6507-1. The hardness was measured five times to check the reproducibility. From the measured data, average hardness has been calculated. The results are shown in Table 5. The measured hardness corresponds to the microstructure types observed in the respective states of the heat-treated C45 carbon steel (Figure 2). The soft-annealed and normalized states were formed by pearlite and ferrite (Figure 2a,b). The amount of ferrite in the normalized state was higher compared to the soft-annealed state. The pearlite has a higher hardness compared to ferrite [47]. The normalized state had a higher hardness compared to the soft-annealed state due to the smallest grain size and lamellar morphology of the pearlite.

**Table 5.** Measured Vickers hardness values of the analyzed samples after heat treatment.

| Sample Heat Treatment | Average Value and Standard Deviation |
|---|---|
| Soft annealed | 207 ± 2 HV 10 |
| Normalized | 273 ± 1 HV 10 |
| Hardened | 694 ± 9 HV 10 |
| Quenched and tempered at 530 °C | 392 ± 8 HV 10 |
| Quenched and tempered at 660 °C | 276 ± 3 HV 10 |

The microstructure of the hardened state consists of martensite, retained austenite, and a small amount of troostite (very fine lamellar pearlite, Figure 2c). Martensite has a high hardness of 700–1000 HV [47,48]. The hardness of retained austenite is also relatively high [48,49]. Therefore, these two microstructure constituents contributed significantly to the observed high hardness of the hardened state of the C45 steel (694 ± 9 HV10, Table 5).

The quenched and tempered at 530 °C state consists of tempered martensite and precipitated cementite particles (low tempered martensite). Although both martensite and cementite are hard, the tempering makes them softer [50]. Therefore, the hardness of the quenched and tempered state has been decreased compared to the hardened state. After quenching and tempering at 660 °C (Figure 2e) the microstructure consisted of sorbite (a very fine mixture of ferrite and cementite particles). Since ferrite is softer compared to both cementite and tempered martensite, the hardness of the material decreased substantially.

*3.3. Roughness, Cylindricality, and Diameter Size Changes*

The heat-treated and machined surfaces were evaluated for geometrical characteristics. The average surfaces roughnesses are shown in Table 6. Table 7 shows the measured values of cylindricality and depicts differences in circularity (diameter sizes) of the machined surfaces. Diameter size changes are calculated relative to 25 mm since the ideal machined diameter size of the machined surface would be 25 mm.

**Table 6.** Measured roughness parameters of heat treated C45.

| Heat Treatment | Machined Surface on Length 30 mm | | | | Machined Surface on Length 40 mm | | | |
|---|---|---|---|---|---|---|---|---|
| | Ra [μm] | Rq [μm] | Rz [μm] | Rt [μm] | Ra [μm] | Rq [μm] | Rz [μm] | Rt [μm] |
| soft annealed | 2.2021 | 2.4087 | 7.8695 | 8.4996 | 2.0789 | 2.2797 | 7.6906 | 8.2754 |
| normalized | 1.8079 | 2.1284 | 8.0620 | 9.2035 | 1.8051 | 2.1171 | 7.8432 | 8.4968 |
| hardened | 1.8871 | 2.2089 | 8.9114 | 10.8660 | 1.7917 | 2.0732 | 8.3776 | 11.3595 |
| QT 530 °C | 2.2973 | 2.5475 | 8.1361 | 8.5743 | 2.3087 | 2.5678 | 8.1069 | 8.5090 |
| QT 660 °C | 1.5912 | 1.9758 | 8.7929 | 11.1379 | 2.2992 | 2.5853 | 9.3118 | 12.1524 |

**Table 7.** Cylindricity and difference in diameter size of machined C45 steel surfaces.

| | Cylindricity on Machined Length [mm] | | Difference in Diameter Size on Machined Length [mm] | |
|---|---|---|---|---|
| | 30 mm | 40 mm | 30 mm | 40 mm |
| soft annealed | 0.01505 | 0.01888 | −0.02058 | −0.01888 |
| normalized | 0.01113 | 0.01025 | −0.00910 | −0.01765 |
| hardened | 0.12679 | 0.17995 | 0.53656 | 0.17338 |
| QT 530 °C | 0.01252 | 0.01440 | 0.03549 | 0.02684 |
| QT 660 °C | 0.01216 | 0.01944 | 0.00218 | 0.01532 |

Figure 3 shows graphically the change in the roughness parameter depending on the C45 steel state. Increased values of some measured values are visible for samples after hardening, quenching, and tempering.

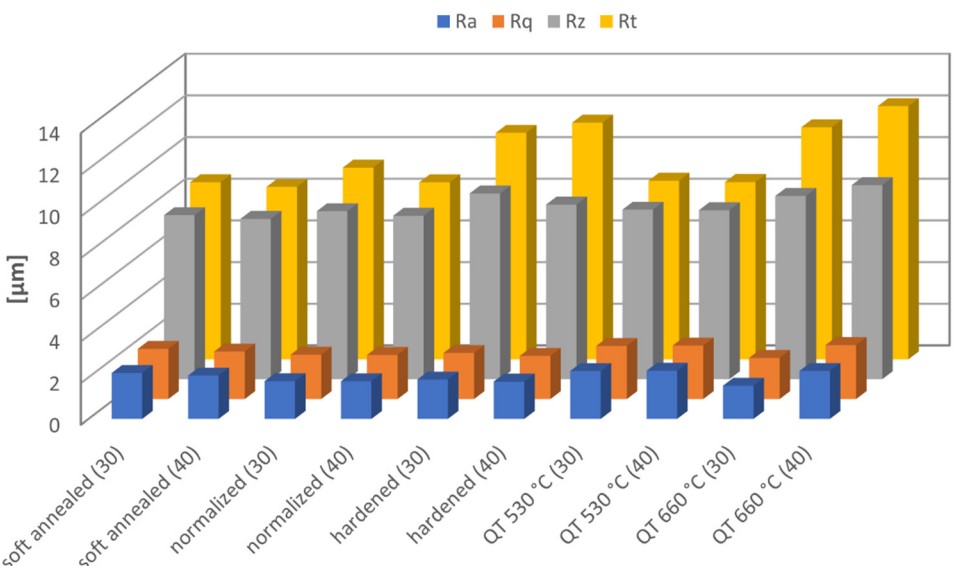

**Figure 3.** Comparison of roughness parameters on all machined surfaces.

A detailed view of the Ra roughness parameter is provided in Figure 4. It shows that the roughness was similar for all analyzed samples, but higher values were observed in the soft annealed state, quenched, and tempered state at 530 °C and on the longer machined surface in the quenched and tempered state at 660 °C. At a soft annealed state, a micro-built-up edge formation [51] can be expected due to a very low hardness (207 HV 10). It can also locally change the cutting wedge geometry and roughness. Quenching and tempering both improve the mechanical properties of the C45 steel, but changes in the microstructure can also be influenced by the selected machining parameters and cutting conditions used due to the presence of tempered martensite lath. It may have led to a possible formation of new phases based on carbides.

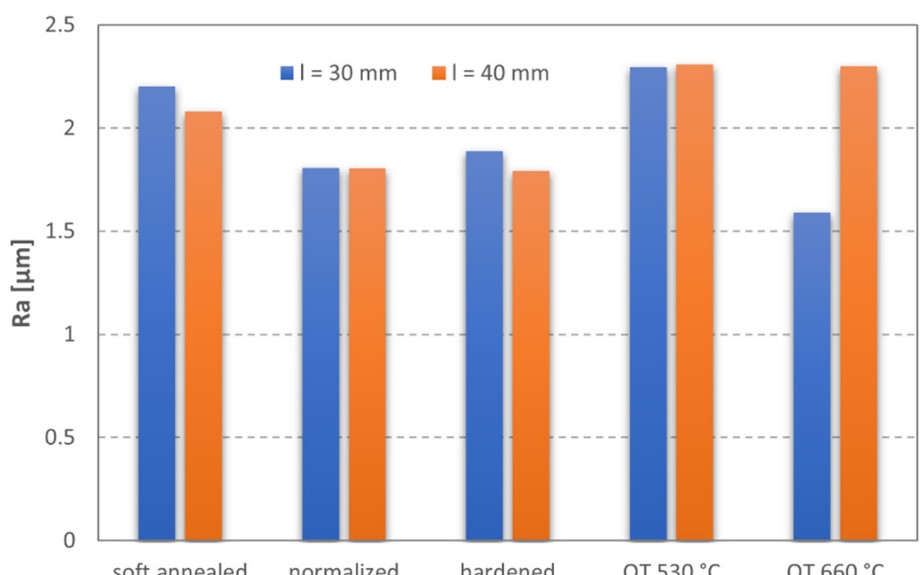

**Figure 4.** Comparison of the Ra parameter for the analyzed states of C45 steel.

Zamrzly studied the turning process of different materials, including the C45 steel [52]. The highest roughness was found for the C45 steel (5.87 µm). The values were decreasing with decreasing machining speed [52]. Therefore, if the lowest surface roughness is required for the design of the technological process of a given workpiece, the lowest possible feed rates and high cutting speeds should be used.

Figure 5 compares the measured values of the Rz roughness parameter for all states of analyzed C45 steel. The maximum values were found for the sample after quenching and tempering at 660 °C and after hardening.

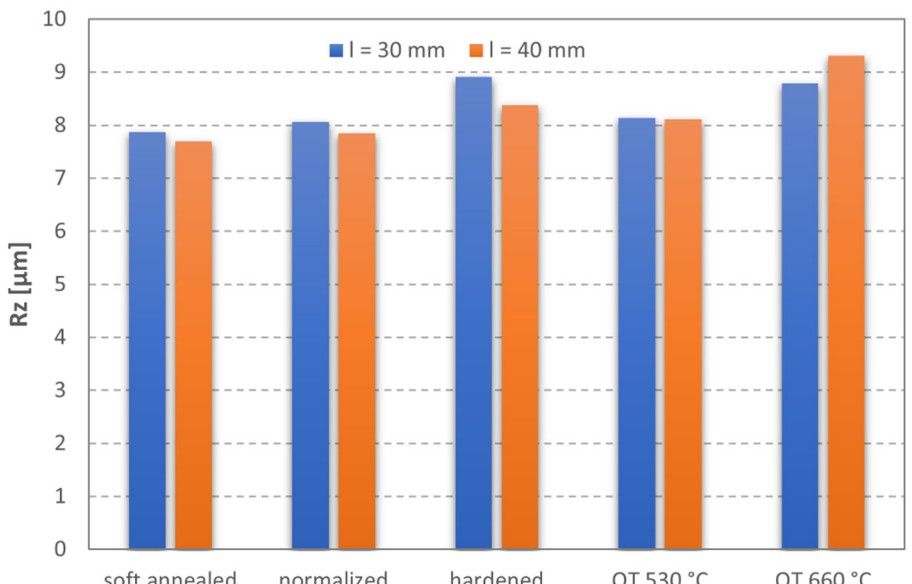

**Figure 5.** Comparison of the Rz parameter for the analyzed states of C45 steel.

The cylindricity of the machined surfaces was measured on each steel. The measured values are shown in Figure 6. The cylindricity was comparable for most steels, however, an exception was found for the sample after hardening. This steel state had a remarkably high value of cylindricity. Similar results were also found for the differences in roundness (circularity). The diameter sizes are compared in Figure 7. A deformation of the cutting edge after turning of 30 mm length was probably smaller than the deformation of the cutting edge of the tool during machining of a longer part of the hardened sample. Samples after soft annealing and normalizing had a negative difference in diameter size. This difference could have been caused by a higher portion of the soft phase (ferrite) which reacted with the tool during turning.

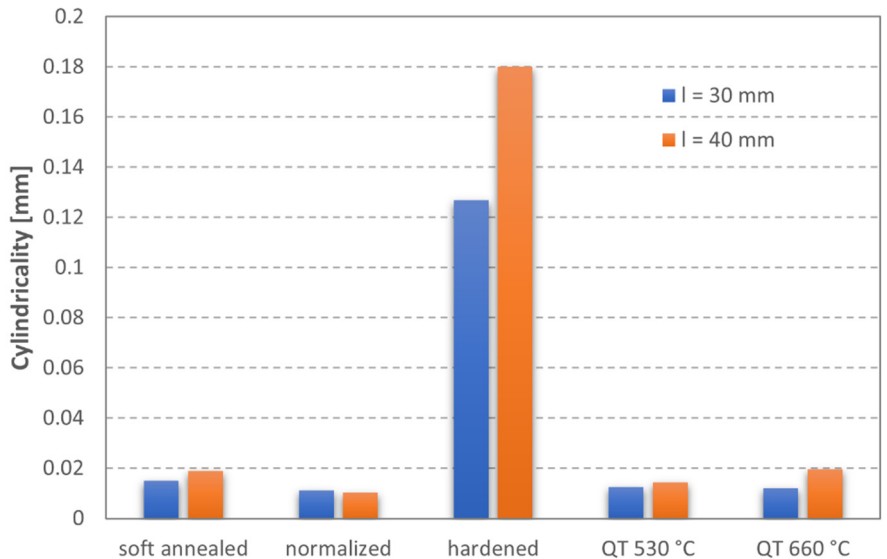

**Figure 6.** Cylindricity of machined surfaces.

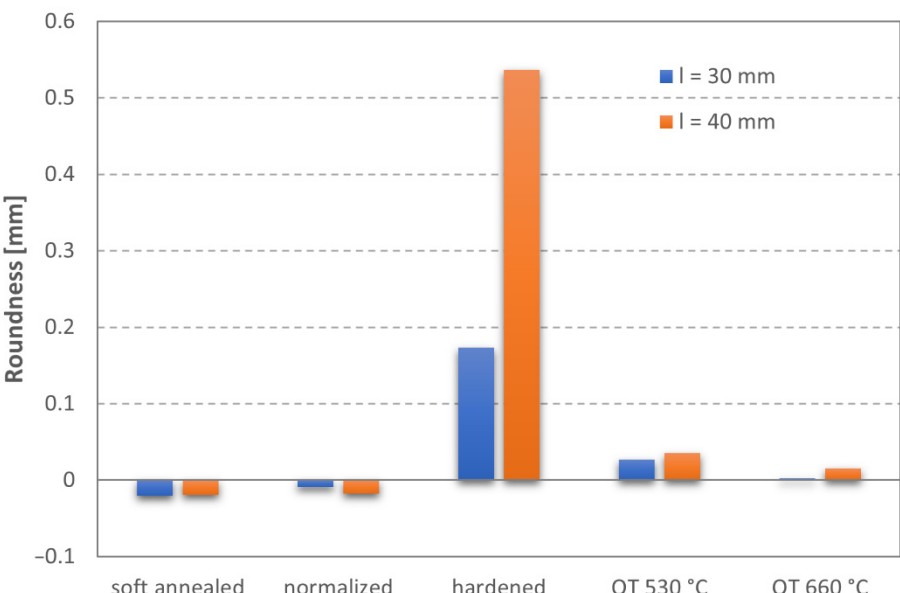

**Figure 7.** Difference in the diameter size of machined surfaces.

The used cutting inserts and their cutting edges were visually inspected after turning. A significant damage to the cutting tip of the cutting insert used was observed after turning of the hardened sample, which is documented in Figure 8. The depth of cutting tip wear was 2.5 mm and the width was approximately 2 mm. For comparison, the cutting insert used for turning of normalizing annealed sample is shown in Figure 9. The cutting edges had no visible damage, except for small areas, probably of a ferrous oxide layer, on major and minor flanks and on the cutting edge tool of the cutting insert used.

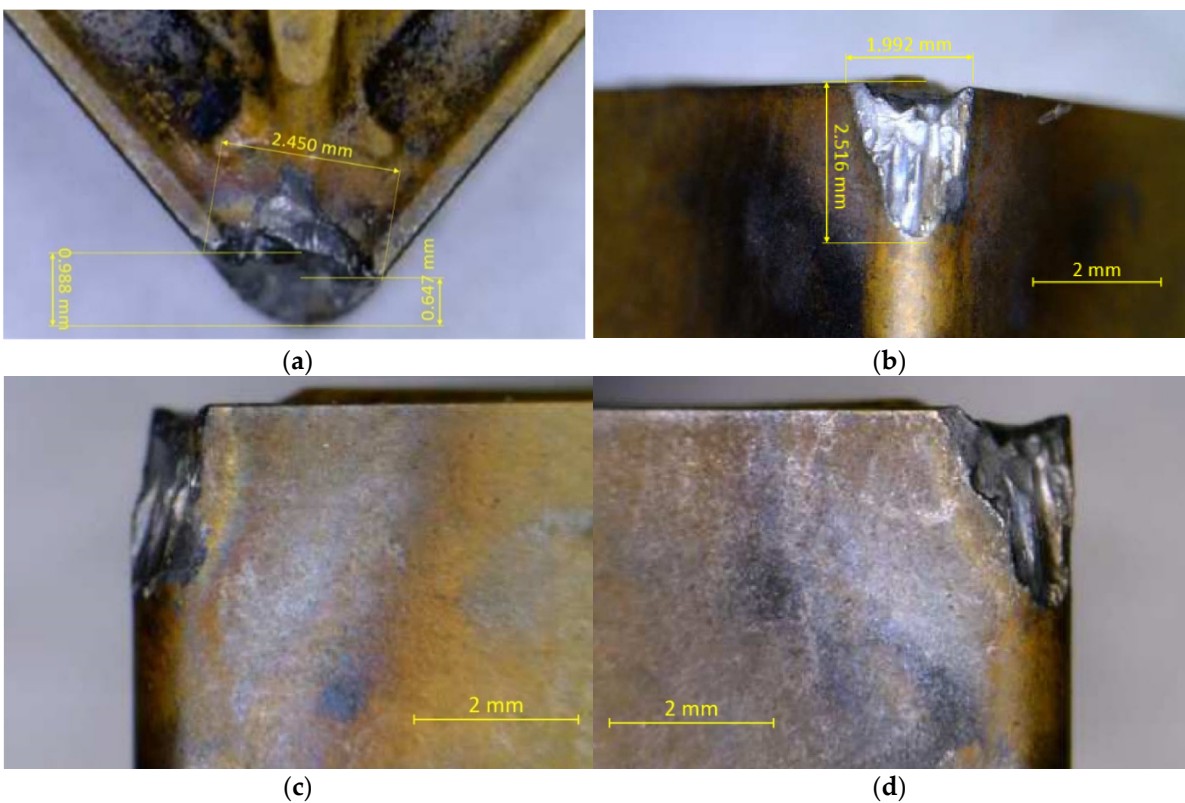

**Figure 8.** Cutting edges after machining the hardened samples. (**a**) wear of the cutting tip; (**b**) depth of wear of the cutting tip; (**c**) view to the major flank; (**d**) view to the minor flank.

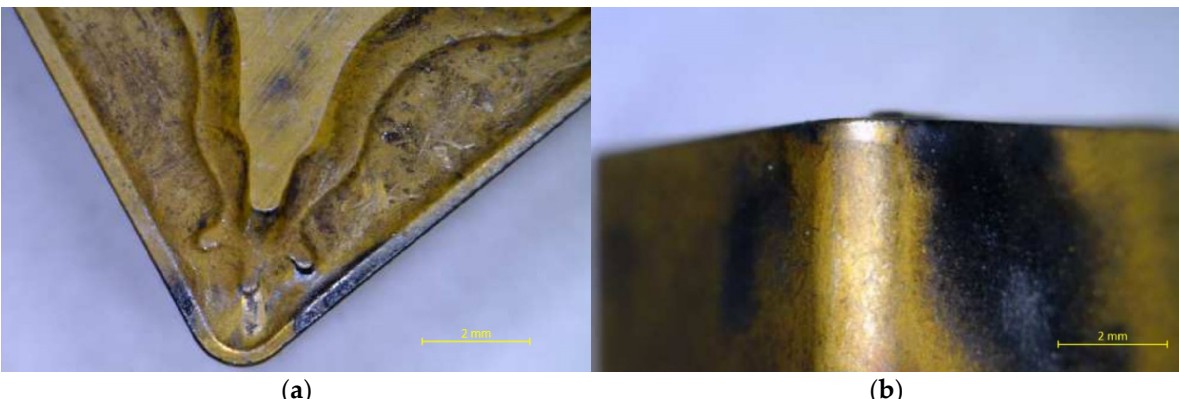

(**a**)             (**b**)

**Figure 9.** Cutting edges after machining of the annealed samples. (**a**) influenced face of the insert; (**b**) major and minor flank influences on the insert.

Martensite present in the hardened specimen had an adverse effect on both cylindricty and roundness (Figures 6 and 7). A similar observation has been also found by Ovali et al. [53]. The authors studied the cylindricity and circularity (roundness) of a gray cast iron. Their samples were austenitized at 900 °C and quenched to salt bath at austempering temperatures 315 °C and 375 °C for various austempering times. Some specimens were also subjected to specific heat treatment to demonstrate the effect of martensite phase on surface topography parameters [53]. All samples were machined at constant feed rate and after heat treatments. The cylindricity of the machined surface increased with increasing temperatures.

Martensite as a hard phase had a negative effect on the surface properties of all heat treated parts (Figure 8). Previous authors studied the effects of heat treatments on the machinability of mild steel [54]. The heat treatment operations applied did not bring about a considerable difference in cutting forces. Therefore, a correlation between the machinability and the hardness of specimens has not been found. Nevertheless, the inspection of the machined surface in [53] revealed that the austenitic-ferritic microstructure improved the surface topography parameters significantly. As such, it can be suggested that the surface topology can be improved by austempering heat treatments. Additional surface treatments are not necessary.

## 4. Conclusions

Heat treatment of non-alloyed carbon steels leads to a change in the material properties. For optimal machinability of steel components, it is necessary to know the initial state of the semifinished product. From the results of our analysis, it can be stated that the machinability of C45 carbon steel was influenced by the initial state. The hardness of the treated steel is very important for the machining process. The article has analyzed the influence of the heat treatment of round C45 steel bar with an initial diameter of 30 mm on the resulting parameters after turning. All heat-treated samples were machined under the same conditions on both sides of the steel bar on two different lengths.

The following conclusions have been made:

1. The soft annealed steel consisted of a soft ferrite matrix. The results obtained show that the diameter change achieved a negative value (−0.03 mm), but the cylindricality was relatively high, ~0.018 mm compared to other samples (beside hardened sample). The built-up edge formation can be explained by the roughness parameter Ra, which was higher compared to other steel states (besides quenched and tempered states). Similarly, a negative value of the diameter size change was observed in the sample in the normalized state, which had the smallest cylindricality value.

2. The highest hardness was found for the steel after the hardening process (694 HV 10). The high hardness was related to the presence of martensite and retained austenite. This type of microstructure caused a damage to the cutting edge of the cut insert

used and led to a significant change in geometrical accuracy. The cylindricity change achieved 0.12 ÷ 0.18 mm compared to the theoretical value of the diameter of the machined steel bar. An inaccuracy was also observed in diameter dimension, on lower length ~0.18 mm and on the second machined surface (machined 40 mm length) a ~0.53 mm difference between the theoretical value of the machined diameter 25 mm was found. The inaccuracies were caused by the wear of the cutting edge of the tool used in cutting parameters setting.

3.  Quenched and tempered states had similar values of the Ra parameter and cylindricality. The Rz parameter for sample QT 530 °C was slightly lower than for QT 660 °C.

4.  The measured roughness parameters are influenced by local cutting conditions and the geometry of the tool used. For the correct cutting parameters evaluation, the initial state of the C45 steel need to be known.

**Author Contributions:** Conceptualization, J.M., R.M. and M.P.; methodology, J.M.; validation, R.M.; investigation, J.M. and R.M.; resources, J.M., R.M. and M.P.; writing—original draft preparation, J.M.; writing—review and editing, R.M. and M.P. All authors have read and agreed to the published version of the manuscript.

**Funding:** This research was funded by the Cultural and Educational Grant Agency of the Ministry of Education. Science, Research and Sport of the Slovak Republic grant No. KEGA 006STU-4/2020 and the Slovak Research and Development Agency project No. APVV-20-0124.

**Institutional Review Board Statement:** Not applicable.

**Informed Consent Statement:** Not applicable.

**Data Availability Statement:** Not applicable.

**Conflicts of Interest:** The authors declare no conflict of interest.

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
