# Peer review of "Effect of Heat Treatment on the Resulting Dimensional Characteristics of the C45 Carbon Steel after Turning"

_metals, doi:10.3390/met12111899_

Round 1

Reviewer 1 Report

Dear authors Your paper is very well written. Although the material is very well known and in many fields some results are known however the paper is very interesting. The paper I very well organized and after change of some minor faults will be suitable for publication. Below I will list some elements that in my opinion have to be made.

1.       Look on references (like line 43) I think there is too many in one gap.

2.       Table 1. Present true chemical composition (good would be to enlarge the present table).

3.       Figure 2. Good would be to present also higher magnification and mark discussed microstructure.

4.       Fig. 3 . change the background.

5.       Conclusions in this version of paper are more a discussion and in text is lack of it. So please change it. Add conclusions in condensed and point form.        

Reviewer 2 Report

The manuscript entitled “metals-1997617-HT” dealing with Heat treatment has been reviewed. The paper has been nicely written but needs significant improvement. Please follow my comments.

1.     The abstract needs to be improved by adding some statistical numbers.

2.     Please double check the chemical composition presented in Table 1. Make sure all are reported correctly. Chemical composition affects the heat treatment and the results.  

3.     Please add the contribution of the paper to the abstract. It is not clear what is the novelty of this work.

4.     How did you select different heat treatment scenarios? Explain it in the text close to table 2.

5.     Please mention what was the gap in research and add a statement to the introduction.

6.     Heat treatment has usage in different applications, especially in additive manufacturing. Authors are encouraged to read and add a short note about the heat treatment in Additive manufacturing by adding the following papers. This robust the quality and contribution of the paper.

·        Effect of post-treatment on local mechanical properties of additively manufactured impellers made of maraging steel

·       A comprehensive review on surface quality improvement methods for additively manufactured parts

·       A review of Industry 4.0 and additive manufacturing synergy

·       Evolution of Microstructure, Texture and Corrosion Properties of Additively Manufactured AlSi10Mg Alloy Subjected to Equal Channel Angular Pressing (ECAP)

Round 2

Reviewer 2 Report

The paper is ready to publish.